# Peripheral Gonadotropin-Inhibitory Hormone (GnIH) Acting as a Novel Modulator Involved in Hyperphagia-Induced Obesity and Associated Disorders of Metabolism in an In Vivo Female Piglet Model

**DOI:** 10.3390/ijms232213956

**Published:** 2022-11-12

**Authors:** Lei Chen, Xin Zhang, Xingxing Song, Dongyang Han, Kaiou Han, Wenhao Xu, Rongrong Luo, Yajie Cao, Yan Shi, Chengcheng Liu, Changlin Xu, Zixin Li, Yinan Li, Xun Li

**Affiliations:** 1College of Animal Science and Technology, Guangxi University, Nanning 530004, China; 2Guangxi Zhuang Autonomous Region Engineering Research Center of Veterinary Biologics, Nanning 530004, China; 3Guangxi Key Laboratory of Animal Reproduction, Breeding and Disease Control, Nanning 530004, China; 4Guangxi Colleges and Universities Key Laboratory of Prevention and Control for Animal Disease, Nanning 530004, China

**Keywords:** gonadotropin-inhibitory hormone (GnIH), feeding behavior, obesity, glucolipid metabolism disorder, insulin resistance, lipid deposition

## Abstract

Apart from the well-established role of the gonadotropin-inhibitory hormone (GnIH) in the regulation of the reproductive functions, much less is known about the peripheral role of the GnIH and its receptor in the metabolic processes. On account of pig being an excellent model for studies of food intake and obesity in humans, we investigated the peripheral effects of the GnIH on food intake and energy homeostasis and revealed the underlying mechanism(s) in female piglets in vivo. Compared to the vehicle-treated group, intraperitoneally injected GnIH significantly increased the food intake and altered the meal microstructure both in the fasting and ad libitum female piglet. GnIH-triggered hyperphagia induced female piglet obesity and altered islet hormone secretion in the pancreas, accompanied with dyslipidemia and hyperglycemia. Interestingly, GnIH decreased the glucose transport capacity and glycogen synthesis, whereas it increased the gluconeogenesis in the liver, while it also induced an insulin resistance in white adipose tissue (WAT) via inhibiting the activity of AKT-GSK3-β signaling. In terms of the lipid metabolism, GnIH reduced the oxidation of fatty acids, whereas the elevated fat synthesis ability in the liver and WAT was developed though the inhibited AMPK phosphorylation. Our findings demonstrate that peripheral GnIH could trigger hyperphagia-induced obesity and an associated glycolipid metabolism disorder in female piglets, suggesting that GnIH may act as a potential therapeutic agent for metabolic syndrome, obesity and diabetes.

## 1. Introduction

Obesity and its related disorders are among the leading causes of illness and mortality in many countries worldwide. The high incidence of obesity is associated with an increased prevalence of insulin resistance and type two diabetes, cardiovascular disease and certain types of cancers [1,2,3]. Notably, accumulating data have been established that demonstrate that prevalent disorders of escalating incidence arise from perturbed neurohormonal pathways or their defective responses to different stressors, and these disorders have a substantial neuroendocrine dimension [3,4,5,6]. Consequently, the neuroendocrine control of obesity and metabolic homeostasis has drawn much attention.

Gonadotrophin-inhibitory hormone (GnIH) was the first avian RFamide peptide identified that directly acts on the pituitary to inhibit a gonadotropin release from the quail hypothalamus via its receptor GPR147 [7,8], and was also subsequently identified in a number of mammals [8,9,10,11]. After over 20 years of research, GnIH has been confirmed to affect the appetite and metabolism of many species, plus it plays a critical role in regulating mammalian reproduction. Tachibana et al. firstly found that GnIH acts as an endogenous orexigen which stimulates feeding behavior in chicks [12]. The consistent results were corroborated in some rodents and birds [13,14,15,16]. Furthermore, it has been established that the accumulation of food intake may lead to the growth of adipose tissue and subsequent obesity and an energy metabolism disorder [17,18]. Thus, the subsequent studies revealed that GnIH increased the food intake and adiposity, whereas it decreased the energy expenditure in mice [19,20]. Recently, our studies have demonstrated that the intraperitoneal administration of GnIH caused the body mass to increase and increased the hyperphagia, hyperlipidemia, hyperglycemia, glucose intolerance, hypoinsulinism, hyperglucagon and insulin resistance in rat and mice [15,16], suggesting that GnIH is a novel neuroendocrine regulator involved in blood glucose homeostasis and may contribute to the occurrence and development of diabetes and obesity. Although a regulatory role of GnIH in appetite and the energy metabolism has emerged, its precise physiological mechanisms remain unknown. In addition, compared to previous rodent species as a model for the study of food intake, obesity and the energy metabolism, pigs, due to their general physiological similarity to humans, offer several advantages and are an excellent model for human studies for the vagal nerve function related to the hormonal regulation of food intake [21]. The similarities between humans and pigs in the organization of the nervous system and digestive tract are well known, while the information obtained from pigs may facilitate an application in humans [22,23].

With over 20 years physical function studies, GnIH acted as a novel neuroendocrine factor, and regulating the feeding and energy metabolism has gradually emerged [15,19,24]. However, unequivocal evidence for the role of GnIH in hyperphagia-induced obesity and the associated metabolism disorders is rare, and the physiological mechanisms by which GnIH affects energy homeostasis, especially lipid homeostasis, remain unknown. In this context, the piglet model is used for the first time in this study to explore the peripheral effects of GnIH on food intake and energy homeostasis, and further revealed the underlying mechanism(s) of GnIH as a metabolic regulator action in peripheral organs including the pancreas, liver and white adipose tissue (WAT) in female piglets in vivo, providing a theoretical basis for the neuroendocrine network to participate in the regulation of the energy metabolism.

## 2. Results

### 2.1. Intraperitoneally Injected GnIH Increases Piglets Food Intake and Alters Meal Microstructure

The effects of acute intraperitoneally injected GnIH on the structure of the first meal in fasting piglets were initially investigated in our food intake analysis. As shown in Figure 1A, GnIH showed that a dose-dependent manner significantly increased the food intake, duration and eating rate of the first meal (*p* < 0.05 and *p* < 0.01) compared to the vehicle-treated group.

To determine whether light played a key role in the GnIH-mediated increase in the food intake of piglets, we further evaluated the effect of an intraperitoneal injection of GnIH on the food intake and meal microstructure of piglets during photophase and scotophase. As shown in Figure 1B,C, 1 mg/mL of GnIH significantly increased (*p* < 0.001) the cumulative food intake during 1 h of the photophase and 3–12 h of the scotophase, while 0.1 mg/mL only markedly increased (*p* < 0.001) the cumulative food intake during 3–12 h of the photophase and scotophase. Furthermore, 0.1 mg/mL and 1 mg/mL of GnIH significantly stimulated (*p* < 0.05, *p* < 0.01 and *p* < 0.001) the meal frequency of the piglets during any period of the photophase and scotophase. Notably, the values obtained for the meal microstructure parameters of chronic GnIH-injected piglets showed (Table 1) that intraperitoneally injected GnIH dramatically increased the meal frequency (*p* < 0.001) and the time spent at meals (*p* < 0.05, *p* < 0.01 and *p* < 0.001), whereas the meal size (*p* < 0.05, *p* < 0.01 and *p* < 0.001), meal duration (*p* < 0.001), intermeal interval (*p* < 0.001) and satiety ratio (*p* < 0.001) were significantly reduced during the photophase and scotophase.

### 2.2. Intraperitoneally Injected GnIH Decreases Feed/Gain Ratio and Increases Pig Obesity

The food intake was monitored in chronic GnIH-injected piglets. Our results showed that after 14 days of a GnIH administration, the average daily feed intake (ADF) (*p* < 0.01 and *p* < 0.001) and average daily gain (ADG) (*p* < 0.05 and *p* < 0.01) were dramatically increased in a dose-dependent manner, whereas the feed/gain ratio was markedly decreased (*p* < 0.05) in different GnIH treatment groups compared with the control group (Figure 1D).

As shown in Figure 1E, the two different doses of GnIH notably increased the body size (*p* < 0.05), neck circumference (*p* < 0.05 and *p* < 0.01) and abdominal circumference (*p* < 0.05 and *p* < 0.01) in a dose-dependent manner compared to the vehicle-treated group. The calculation revealed that the POI was dramatically higher (*p* < 0.01) in the GnIH-injected groups than the vehicle-treated group, indicating that obesity occurred in piglets after a GnIH injection.

### 2.3. Intraperitoneally Injected GnIH Influence Serum Biochemical Indexes and Alters Body Fat Composition

To investigate what were the reasons which contributed to the significantly elevated GnIH-induced body mass in piglets, the serum biochemical indexes were measured. As shown in Figure 2A, chronic intraperitoneally injected GnIH in a dose-dependent manner significantly increased the concentrations of TG (*p* < 0.05 and *p* < 0.01), FFA (*p* < 0.05 and *p* < 0.01) and GLU (*p* < 0.05 and *p* < 0.01). The concentration of LDL-C was only increased at the high dose of the GnIH group, whereas no differences were observed in the CHOL, HDL-C, LDH, ALT and AST after a chronic GnIH injection compared with the vehicle-treated piglets. In addition, no differences in the AST/ALT values were calculated between the groups, indicating that all animals did not develop hepatitis after a GnIH administration during the study.

To investigate what were the reasons which contributed to significantly elevated GnIH-induced body mass in piglets, the weight of some organs or tissues were measured. As shown in Figure 2B,C, the mass of iWAT (*p* < 0.05) and pgWAT (*p* < 0.05 and *p* < 0.01) were markedly increased in a dose-dependent manner after the GnIH treatments. In addition, the masses of the liver (*p* < 0.05) and pancreas (*p* < 0.01 and *p* < 0.001) were increased, whereas the masses of the kidney and spleen did not change compared to the vehicle-treated group.

### 2.4. Effects of Intraperitoneally Injected GnIH on Glucose Homeostasis and Islet Histomorphology Changes for Investigated

To investigate the effects of GnIH on glucose homeostasis, the effect of acute and chronic intraperitoneally injected GnIH on the fasting blood glucose levels was first evaluated at different time points in piglets who fasted for 8 h. As shown in Figure 3A, the chronic fasting blood glucose levels dramatically increased (*p* < 0.01 and *p* < 0.001) in a dose-dependent manner for 15 min post-GnIH injection and then gradually decreased from 30 to 60 min after the GnIH injection compared with the vehicle-treated piglets. The acute fasting blood glucose curve is similar to the chronic fasting blood glucose. Interestingly, the fasting blood glucose levels were still markedly higher (*p* < 0.05 and *p* < 0.01) 60 min after the GnIH administration in the chronic GnIH-treated piglets. Furthermore, the initial blood glucose levels (0 min) in the chronic GnIH-injected piglets showed significant increases (*p* < 0.05 and *p* < 0.01) compared with the control piglets, whereas no difference was observed between the acute GnIH-treated and the vehicle-treated groups (Figure 3B). A similar result was observed for the augmented AUCfasting blood glucose in the chronic high dose of the GnIH-treated piglets. However, only the high dose of GnIH in the acute injection group was observed a significant increase (*p* < 0.05) in the AUCfasting blood glucose, compared to the control group.

To investigate the effects of intraperitoneally injected GnIH on glucose elimination, intraperitoneal glucose tolerance tests were performed for the piglets that fasted who were acutely or chronically administered two doses of GnIH. As shown in Figure 3C, the blood glucose concentrations were significantly higher in the GnIH-treated piglets than in the vehicle-injected piglets, as shown by the area under the curve values, although the glucose tolerance curves appeared slightly different between the piglets injected with acute and chronic doses of GnIH. Notably, despite the blood glucose levels peaking at 15 min and then gradually decreasing from 15 to 135 min in all groups, the GnIH induced significant hyperglycemia throughout the duration of the test in piglets treated with high acute and chronic doses of GnIH. In addition, significantly higher glucose levels were detected 15–30 min after the administration of low acute or chronic doses of GnIH compared with the control. Taken together, these results indicate that GnIH inhibited the exogenous glucose eliminate, which was most significant for the piglets who were administered a high chronic dose of GnIH compared with the other treatments.

To elucidate the mechanism of GnIH-induced hyperglycemia in piglets who were chronically administered GnIH, the pancreatic islet morphology as well as the expression of insulin and glucagon were assessed in the pancreas of piglets that were administered chronic doses of GnIH via an intraperitoneal injection. As shown in Figure 3D, although we observed the islet hypertrophy, the analysis of the mean optical density values revealed a significant suppression (*p* < 0.05 and *p* < 0.01) of insulin secretion combined with a significantly elevated (*p* < 0.05 and *p* < 0.01) glucagon secretion following the chronic GnIH treatments. In addition, the *insulin* mRNA expression levels were markedly decreased (*p* < 0.01), whereas the *glucagon* mRNA expression levels were dramatically increased (*p* < 0.01 and *p* < 0.001) in the piglets who were administered different doses of GnIH, consistent with the results of immunohistochemistry. 

### 2.5. Effect of Intraperitoneally Injected GnIH on the Expression Levels of Hepatic Glucometabolism-Related Genes

To elucidate the molecular mechanism underlying the GnIH-mediated glucose metabolism disorders, the expression of glucometabolism-related genes was detected in the liver of piglets administered with chronic doses of GnIH. Compared to the vehicle-treated piglets, the *IRS2* (insulin receptor substrate 2) and *GCK* (glucokinase) mRNA expression levels were significantly diminished (*p* < 0.05 and *p* < 0.01), whereas the *PEPCK* (phosphoenolpyruvate caboxykinase) mRNA expression levels were significantly increased (*p* < 0.05) in the piglets who were injected with high and low doses of GnIH. However, the *GLUT4* (glucose transporter 4) and *IRS1* mRNA expression levels were only significantly decreased (*p* < 0.05) in the high-dose GnIH group. Moreover, no difference was observed in the mRNA expression levels of *FBP1* (fructose-1,6-bisphosphatase 1) after the GnIH injection compared with vehicle-treated piglets (Figure 4A). In addition, we further assessed whether GnIH regulates the insulin-associated AKT-GSK3β signaling cascade in the liver of piglets. The results showed that the GnIH-induced phosphorylation of AKT and GSK3β significantly decreased (*p* < 0.05 and *p* < 0.01) in a dose-dependent manner (Figure 4B). 

### 2.6. Effect of Intraperitoneally Injected GnIH on the Expression Levels of Glucometabolism-Related Genes in WAT

To elucidate the molecular mechanism underlying the GnIH-induced hyperglycemia and insulin resistance, the expression of the insulin resistance-related genes and insulin-associated AKT-GSK3β signaling cascade was detected in the iWAT and pgWAT of chronic GnIH-treated piglets. As shown in Figure 5A,C, a high-dose of GnIH induced notable increases in the *GLUT4*, *IRS1* and *IRS2* mRNA expression levels (*p* < 0.05 and *p* < 0.01) in the iWAT and pgWAT, whereas only a low-dose of GnIH significantly elevated (*p* < 0.05) the expression level of *GLUT4* in the iWAT.

The Western blot results showed that the GnIH-induced phosphorylation of AKT markedly increased (*p* < 0.01 and *p* < 0.001) in a dose-dependent manner in iWAT and pgWAT. In contrast, the GnIH-induced phosphorylation of GSK3-β was significantly decreased (*p* < 0.05 and *p* < 0.01) in the iWAT and pgWAT, compared to the vehicle-treated piglets (Figure 5B,D).

### 2.7. Effect of Intraperitoneally Injected GnIH on the Expression Levels of Hepatic Lipid Metabolism-Related Genes

To reveal the molecular mechanism underlying the GnIH-induced liver mass increase and hyperlipidemia after the chronic GnIH injections in piglets, we subsequently examined the mRNA expression levels of lipid metabolism-related genes in the liver. As shown in Figure 6A, a high dose of GnIH induced a notable increase in the *ACC* (acetyl-CoA carboxylase) and *FABP4* (fatty acid binding protein 4) mRNA expression levels (*p* < 0.05). On the contrary, GnIH markedly decreased (*p* < 0.05 and *p* < 0.01) the expression levels of *CPT-1* (carnitine acyltransferase I), *AMPKα1* (AMP-activated protein kinase α1), *LXRα* (liver X receptorsα) and *ATP-5B* (ATP synthase 5B) in a dose-dependent manner. Simultaneously, no significant difference was observed in the expression levels of *FASN* (fatty acid synthase) and *ACLY* (ATP citrate lyase) after a GnIH injection compared with the vehicle-treated piglets in the liver.

To investigate whether the GnIH regulated the hepatic lipid metabolism via the active AMPK signal pathway, the effect of intraperitoneally injected GnIH on the hepatic AMPK signaling cascade was further evaluated. The results showed that the GnIH-induced phosphorylation of AMPKα and AMPKβ dramatically decreased (*p* < 0.05, *p* < 0.01 and *p* < 0.001). In addition, the GnIH induced a significant increase in the protein expression levels of the C/EBPα (CCAAT-enhancer binding protein α) and the PPARγ (peroxisome proliferator-activated γ) receptor (*p* < 0.05 and *p* < 0.01) (Figure 6B).

### 2.8. Effect of Intraperitoneally Injected GnIH on the Expression Levels of Lipid Metabolism-Related Genes in WAT

Since white adipose tissue plays an important role in maintaining the balance of the lipid metabolism in the body, the effect of intraperitoneally injected GnIH on the expression of the lipid metabolism-related genes and the activation of the AMPK signaling cascade was detected in the pgWAT and iWAT. As shown in Figure 7A,B, the GnIH induced a marked increase in the mRNA expression levels of *ACC*, *FASN* and *HSL* (hormone-sensitive lipase) in the iWAT and pgWAT (*p* < 0.05 and *p* < 0.01), whereas the expression levels of *CPT-1* and *AMPKα1* were significantly decreased (*p* < 0.05, *p* < 0.01 and *p* < 0.001) in a dose-dependent manner. Simultaneously, no significant difference was observed in the expression levels of *FABP4* after the GnIH injection compared with the vehicle-treated piglets in iWAT.

The Western blot results showed that GnIH notably inhibited (*p* < 0.05, *p* < 0.01 and *p* < 0.001) the phosphorylation of AMPKα in a dose-dependent manner both in the iWAT and pgWAT. Although a high dose of GnIH induced significant reductions in the phosphorylation of AMPKβ (*p* < 0.01) in the iWAT and pgWAT, only the low-dose treatment of GnIH showed a markedly suppressed (*p* < 0.01) AMPKβ phosphorylation in the iWAT. In contrast, all doses of GnIH induced significantly higher protein expression levels of C/EBPα, PPARγ and ACLY (*p* < 0.05 and *p* < 0.01) both in the iWAT and pgWAT (Figure 7C,D).

## 3. Discussion

Ever since GnIH was reported as an innate hunger factor in chicks [12,25], a similar result that GnIH elevated the cumulative food intake was also observed in sheep, mice and cynomolgus monkeys via an ICV infusion [26]. Recently, our newest study first confirmed that the effect of acute and chronic intraperitoneally injected GnIH on food intake was similar to that of an ICV injection in rats and mice [15,16]. Now, the present study validated the effect of peripherally treated GnIH on the appetite of both fasting and ad libitum-fed piglets, suggesting that central or peripheral GnIH equally act as conservative endogenous orexigens in vertebrates. Our previous studies indicated that the light cycle plays a crucial role in the GnIH-mediated increase in the food intake in rodents, but the effect of intraperitoneally injected GnIH on food intake seems different during different periods of the photophase and scotophase post-GnIH injection in different species. The results of our study showed that different doses of GnIH both induced significant increases in the food intake during the scotophase, especially within the 3 to 12 h period of the scotophase, whereas only a high dose of GnIH markedly elevated the food intake during the photophase, especially in the first h of the photophase post-GnIH injection in ad libitum-fed piglets. We presumed that the differences in the natural preference of food ingestion during the photoperiod or a sensitivity to the light-mediated food intake between rodents and pigs contributed to the different effects on the GnIH-induced food intake accumulation during different periods of the photophase and scotophase in different species. Although this result combined our previous findings which suggested that the light cycle affected the GnIH-induced food intake accumulation, the relationship between the light perception and GnIH is more complicated than our investigation suggests, and this issue needs further study in more mammals in the future. The analysis of the meal pattern is of primary importance to assess the mechanisms regulating the feeding behavior. We observed that GnIH markedly elevated the meal duration and eating rate of the first meal in fasting piglets, while significantly increasing the meal frequency and time spent in meals, reducing the meal size, the inter-meal interval and the meal duration in ad libitum-fed piglets whenever during the period of the photophase or scotophase, consistent with the results of the meal microstructure analysis in intraperitoneally GnIH-treated rats and mice [15,16]. Notably, the satiety ratio was decreased after the intraperitoneal injection of GnIH, suggesting that the exogenous activation of the GnIH pathways impairs the satiety value of the food ingested during spontaneous nocturnal feeding. These data corroborate the previous suggestion that intraperitoneally injected GnIH increased the food intake by causing changes in the meal’s microstructure, as supported by the observed reduction in satiety and as influenced by the light.

The increased food intake, especially feeding at night, and changes in the meal’s microstructure increases the fat deposition and leads to an increased risk of obesity [27,28,29]. Our previous study determined that intraperitoneally injected GnIH not only increased the food intake and altered the meal’s microstructure but also increased obesity in rats and mice [15,16]. Similar results were found in our present body mass investigation that GnIH triggered an increase in the average daily intake, resulting in an average daily gain and an elevated POI while the feed/gain ratio decreased, suggesting that chronic intraperitoneally injected GnIH induced hyperphagia, resulting in obesity. Subsequently, the results of the tissue and organ mass investigation showed that the pgWAT and iWAT accumulated, as well as increases in the gains of the liver and pancreas, that may be responsible for the increased body weight in the chronic intraperitoneally GnIH-treated piglets. Our results correspond with the previous study in that the mass of the liver and brown fat mass were increased by the chronic intracerebroventricular infusion of GnIH in lean male C57BL/6J mice, despite the mass of the white adipose tissue being unchanged [20]. Moreover, the concentrations of the serum TG, FFA, LDL-C and glucose were notably increased in the GnIH-treated piglets, which supported the above results of food intake and body and organ mass measurements. The corresponding results were also confirmed in our previous studies in rodents [15,16]. Similarly, Anjum et al. demonstrated that GnIH mediated the increase in the uptake of nutrients (glucose and TGs) in the adipose tissue, resulting in the accumulation of fat and an increased body mass in mice [19]. However, other studies found that although intracerebral injection of GnIH increased the endogenous glucose levels in rats and mice, the serum lipid and insulin levels did not change [20,30]. Possible explanations for the discrepancy to the results are that a central or peripheral administration of GnIH may have a different mechanism of regulating the energy metabolism. In addition, the different treatments protocol or doses may cause different results in the same species. Notably, we observed that intraperitoneally injected GnIH triggers several metabolic abnormalities in piglets, featuring obesity, hyperglycemia, and dyslipidemia that perfectly accorded with the definitions of the metabolic syndrome (MS) given by the World Health Organization (WHO), the National Cholesterol Education Program-Adult Treatment Panel (NCEP-ATP III) and the International Diabetes Federation (IDF). MS can be considered as a combination of metabolic disorders which, when occurring together, increase the risk of developing cardiovascular disease and diabetes. Thus, we hypothesized that GnIH would be a novel neuroendocrine regulator involved in the development and progression of obesity, MS and diabetes.

The clues from the results described above indicate that peripheral GnIH may be involved in hyperphagia-induced obesity and the associated disorders of the energy metabolism, which prompted us to investigate the effect of intraperitoneally injected GnIH on glucose and lipid homeostasis, and further revealed the underlying mechanism(s) of GnIH as a metabolic regulator action in the peripheral organs. The effects of acute and chronic intraperitoneally injected GnIH on the fasting blood glucose levels and glucose elimination in piglets were primary evaluated. Our results are consistent with our previous research on rodents that were intraperitoneally injected with GnIH where we could instantly and markedly observe elevate blood glucose levels within 15 min, while this dramatically inhibited the glucose elimination response both in acute and chronic intraperitoneally GnIH-injected piglets, suggesting that GnIH induced a glucose intolerance and hyperglycemia in piglets. Notably, the long-term chronic GnIH treatment particularly reinforced these effects, which validated our similar findings in rats [15]. A previous study demonstrated that a long-term low glucose tolerance is likely to develop into diabetes, which can be accompanied by hypertension, hyperlipidemia and obesity [31]. Therefore, these results suggested that a GnIH-induced glucose intolerance is one of the factors which promotes hyperlipidemia and obesity, and even the development of diabetes in piglet models. Our previous investigations, and those of other studies, have been confirmed in the way that GnIH and its receptor GPR147 distributed in mouse and porcine pancreatic islets [15,32], suggesting that GnIH could directly act on pancreatic islets to regulate glucose homeostasis. To decipher the molecular mechanism underlying the GnIH-mediated hyperglycemia in piglets, the effects of intraperitoneally injected GnIH on the pancreatic morphology and function were subsequently investigated. We observed that an intraperitoneally injected GnIH chronic intraperitoneal GnIH treatment dramatically promoted pancreatic islet hyperplasia. Concomitantly, the synthesis and secretion of insulin were significantly decreased, but glucagon did the opposite. Our previous study preformed in rats not only found equal results in the pancreatic islet but it also indicated that GnIH induced a β-cell dysfunction and a compensatory increase in the number of α and β cells in islets, which may be responsible for pancreatic islet hyperplasia [15]. Our present and previous studies suggested that GnIH could directly change the pancreatic morphology and glucose homeostasis-related hormone secretion in the pancreas, resulting in hyperglycemia.

The symptoms of chronic intraperitoneally GnIH-treated piglets exhibited hyperglycemia, a glucose intolerance, hypoinsulinism and hyperglucagon, which prompted us to decipher the molecular mechanism underlying the GnIH-mediated insulin resistance in insulin-stimulated glucose disposal tissues. The liver, as an important target organ of insulin, plays a central role in glucose homeostasis because it controls the blood glucose levels through glycogenolysis and gluconeogenesis [33]. The results of our hepatic insulin signal analysis showed that the *GLUT4*, *IRS1* and *IRS2* mRNA expression levels were notably down-regulated, accompanied by a significantly suppressed insulin-associated AKT-GSK-3β signaling cascade after a chronic intraperitoneal GnIH injection in piglets, which is consistent with our studies and those of others performed in rodents [15,19]. It has been reported that insulin binding to insulin receptors requires the phosphorylation of the IRS, which activates the Akt-GSK-3β signaling and further controls the glucose mobilization and transport [34]. An inactive GSK3β-induced complex I dysfunction has been reported to promote the ROS production and contribute to insulin resistance [35,36]. These results suggested that GnIH, though an inhibit glucose absorption and hepatic insulin signal cascade, leads to an increased insulin resistance. Furthermore, GnIH was also observed to promote a significant change in the expression of *PEPCK* (gluconeogenic rate-limiting enzyme) but inhibit the expression of *GCK* (glycolysis rate-limiting enzyme) in the livers, suggesting that GnIH disrupted the hepatic glucose metabolism through suppressed glycolysis and elevated gluconeogenesis. Notably, a previous study indicated that when hepatic insulin signaling is impaired in type 2 diabetes, its ability to induce a glycogen decomposition and gluconeogenesis is impaired, leading to an excessive glucose production and further fasting hyperglycemia in diabetes [37]. Based on these findings, we suggested that GnIH impaired a hepatic glucose transport and an insulin signal transduction, accompanied with hepatic gluconeogenesis and an increased glucose output, but a decreased glycolysis, resulting in sustained extracellular hyperglycemia and an insulin resistance, increasing the risk of diabetes in female piglets. This conclusion was further supported by the reduced expression observed for the insulin receptor and *GLUT4*, as well as the suppression of the GSK-3β signaling in the pgWAT and iWAT of piglets who received chronic intraperitoneally injected GnIH. However, the AKT phosphorylation was significantly increased in the pgWAT and iWAT, which was contrary with that of the liver. Because the activation of AKT is known to facilitate glucose mobilization and transport, we hypothesized that the GnIH-induced glucose absorption increased, resulting in an accumulation in the WAT, which was supported and corresponded with the results of the WAT mass measurement.

Notably, an insulin resistance is considered to be selective, with insulin normally promoting anabolism in the liver by increasing the glucose consumption and lipid synthesis. This activity should be inhibited when the liver becomes insulin resistant, but actually the liver lipid synthesis is increased, leading to hyperglycemia and hypertriglyceridemia [38,39]. Furthermore, we observed that piglets that received an intraperitoneal GnIH injection exhibited obesity and hyperlipidemia along with hyperglycemia, a glucose intolerance and an insulin resistance. These data prompted us to further decipher the molecular mechanism underlying the GnIH-mediated lipid metabolism disorder in the liver and WAT. Interestingly, the communal results were found in the expression of lipid metabolism-related genes and the AMPK signal cascade in the liver, pgWAT and iWAT of piglets. Our results showed that GnIH could significantly increase the expression of *AMPKα1* and adipogenesis-related genes (*ACC* and *FASN*), whereas it reduced the expression of fat oxidation decomposition genes (*CPT-1*) in the liver, pgWAT and iWAT of piglets. In addition, the results of our Western blot analysis were supported above the results that GnIH dramatically inhibited the activation of AMPKα and AMPKβ, whereas the expression of C/EBPα, PPARγ and ACLY were notably elevated in the liver, pgWAT and iWAT of intraperitoneally GnIH-injected piglets. It is well known that AMPK signaling has important implications for the lipid metabolism by directly phosphorylating proteins or modulating the gene transcription in specific tissues, such as in the liver and fat [40,41]. An AMPK activation negatively regulates the mTOR/PPARγ/CCAAT-enhancer binding protein (C/EBP) α cascade and downstream ACC to produce ATP, thereby reducing the fatty acid synthesis [42,43,44]. ACC is an important site of regulation within the fatty acid synthesis and oxidation pathways, as it can not only allosterically inhibit CPT-1, a key enzyme for β-oxidation, but also catalyze the main substrate for FASN to the de novo synthesis of long-chain fatty acids [40,45]. Based on the above data, we suggested that GnIH promoted fatty acid synthesis but inhibited a fat oxidation by inhibiting the AMPK signaling pathway in the liver, pgWAT and iWAT, eventually leading to obesity-related lipid metabolism disorders in piglets. Furthermore, previous studies have confirmed that the HSL is related to the increase in lipolysis [40], while the decrease in LXRα may lead to a disorder of the cholesterol metabolism and the increase in the peripheral blood glucose [46]. Therefore, this hypothesis was also supported by the reduced expression observed for *LXRα* and *ATP-5B* in the liver, as well as the increased expression observed for *HSL* in the white adipose tissue. This is the first piece of evidence which revealed that the molecular mechanism of GnIH is involved in the lipid metabolism in peripheral organs and tissues, suggesting that intraperitoneally injected GnIH promoted adipogenesis and decreased lipolysis in the liver and WAT, resulting in a lipid metabolism disorder, exhibiting hyperlipidemia and obesity in piglets. However, the fully molecular mechanism of GnIH regulating the lipid metabolism is more complicated than we have revealed in the present study. Specifically, our vivo study cannot reveal the direct effects and precise mechanism of GnIH on the lipid metabolism in the liver and WAT. This limitation will conduct us to design some in vitro experiments for further studies.

In summary, the results of this study demonstrated that peripheral GnIH plays a critical regulatory role on food intake and the energy metabolism. The molecular mechanism of a GnIH-induced glucose metabolic disorder revealed that GnIH impaired the glucose homeostasis via the affect the AKT-GSK-3β signaling cascade had to alter the function and morphology of the pancreas islets, impair the glucose transport and cause an insulin signal transduction in the liver and WAT (Figure 8). Interestingly, we found that peripheral GnIH could directly act on peripheral organs, such as the liver and WAT, to impair lipid homeostasis in piglets. The possible molecular mechanism is that intraperitoneally injected GnIH promoted adipogenesis and decreased lipolysis in the liver and WAT via inhibiting the AMPK signal pathway, resulting in hyperlipidemia and obesity (Figure 8). To our knowledge, this is the first study which systematically provided the phenotype of intraperitoneal GnIH-treated piglets and further elucidated the mechanism of peripheral GnIH-induced obesity and associated glucose and lipid metabolism disorders in an in vivo female piglet model, suggesting that GnIH as a novel neuroendocrine regulator, GnIH agonist and GnIH antagonist may provide potential therapeutic properties for obesity, as well as its related disorders such as MS, diabetes and even cardiovascular disease. Notwithstanding this, our understanding of the physiological mechanisms of the GnIH in the energy metabolism is still inconclusive. Future studies are urgently required to reveal the roles of GnIH on the development and progression of obesity and the related metabolic diseases. Furthermore, the expression of GnIH and its receptor in different metabolic disorder models deserves to be investigated in future studies.

## 4. Materials and Methods

### 4.1. Animals and GnIH

All of the experiments were performed in accordance with the guidelines of the regional Animal Ethics Committee and were approved by the Guangxi University Ethical Committee (Project ID GXU2020-089). Eighty-eight crossbred Large White-Landrace × Yorkshire × Duroc purebred piglets (approximately 5 weeks old, of similar size and mean BW was 14.5 ± 0.2 kg), with no previous evidence of health problems and adequate pathogen monitoring reports, were used. Immediately after being weighed, the piglets were placed in individual small metal wire cages. The piglets were maintained under constant conditions of light (12:12 light–dark cycle) and temperature (24–26 °C), with free access to grow pig compound feed (piggery1#, New Hope Liuhe, Chengdu, China) and water.

Human GnIH ortholog (catalog No. 048-46, Phoenix Pharmaceuticals, Burlingame, CA, USA) was used in the present study. The amino acid sequences of human GnIH coincided with the pig GnIH sequences (Val-Pro-Asn-Leu-Pro-Gln-Arg-Phe-NH2).

### 4.2. Animal Treatments

#### 4.2.1. Grouping

Female pigs were acclimatized for 7 days after purchase and randomly divided into 3 groups (10 pigs in each group), including the control group (1 mL saline/each), the GnIH low dose group (0.1 mg/1 mL of GnIH, 1 mL/each) and the GnIH high dose group (1 mg/1 mL of GnIH, 1 mL/each). GnIH was dissolved in a 0.9% saline solution. The concentrations of GnIH used in the present study were decided according to our pre-experiments and previous studies [15,16]. In addition, half of each group (*n* = 5 per group) were randomly selected for the acute intraperitoneal injection trials and the other half (*n* = 5 per group) for the chronic intraperitoneal injection trials.

#### 4.2.2. Injections Protocol

An acute intraperitoneal injection was defined as piglets who received only one injection of different doses of GnIH or saline. In terms of the chronic intraperitoneal injection, it was defined as piglets who received two intraperitoneal injections (7:00 a.m. and 7:00 p.m.) of different doses of GnIH or saline for 14 consecutive days, respectively.

#### 4.2.3. Fasting Treatments

Before the measurements of the first post-fasting meal microstructure, fasting blood glucose and oral glucose tolerance tests, the pigs in each group were fasted for 8 h before the tests but without restricting their water intake.

### 4.3. Food Intake, Meal Microstructure and Weight Measurements

After the fasting treatments, acute intraperitoneally GnIH-injected piglets were used to evaluate the structure of the first post-fasting meal microstructure, including the feeding volume (g), feeding duration (min) and feeding rate (feeding volume/feeding time) at 7:00 a.m. The feeding volume was defined as the feed provided before the start of the trial (5 kg/each) minus the feed remaining after the trial. The duration of the feeding was recorded from the start of the piglet’s feeding until it stopped after a period of more than 10 min of non-feeding, as previously shown.

Piglets that received chronic intraperitoneal injections of GnIH were used for the food intake and weight measurements. After the injection, the piglets were placed back in the cages with a known weight of the compound feed. The individual body weight and feed intake were measured each day at 7:00 a.m. for 14 consecutive days. The average daily gain (ADG) was calculated as the total weight change divided by the number of days. The average daily feed intake (ADFI) was calculated as the total food intake divided by the number of days, and the feed/gain (F/G) was calculated as the ADFI divided by the ADG.

In the meal microstructure of the ad libitum feeding test, piglets who received chronic intraperitoneal injections of GnIH were performed under light and dark environments, and the segmented cumulative feeding volume (g), feeding times (times) and feeding duration (min) were recorded at 1 h, 2 h and 3–12 h after the injection. At the same time, the feeding behaviors of the pigs were observed and calculated, including the feeding rate (feeding volume/feeding time), feeding time per meal [(feeding time/feeding (number of times)], feeding interval [(total duration of recording time-total duration of feeding time)/number of feeding times] and satiety rate (average feeding interval/average amount of food taken per meal). The log-survivor technique described by Bigelow and Houpt (1988) was used to analyze all the feeding behaviors, with a standard value of 2 min for the feeding defined. The amount of food taken per meal was more than 10 g. When the interval between two consecutive visits was shorter than 2 min, they were considered to belong to the same meal [47,48].

### 4.4. Porcine Obesity Index

According to the method of measuring and calculating the porcine obesity index (POI) of pigs proposed by Sebert [49], the anthropometric parameters such as the body size and abdominal and neck circumferences were measured of the chronic intraperitoneally GnIH-injected piglets before their slaughter. The calculation formula was as follows: POI = 1000π(BS/3)(A2 + N2 + A × N)/BS. In the formula, the base is represented by the abdomen (A), the top by the neck (N) and the length by the body size (BS), while the BS is the body size and A and N are the radiuses of the abdomen (A) and the neck (N).

### 4.5. Serum and Tissue Collection

The serum and tissues from the piglets having undergone chronic intraperitoneal GnIH or saline administration for 14 days were collected. Briefly, the bloods were obtained by the jugular vein puncture 15 min after the last GnIH injection, coagulated at room temperature for 2 h after the collection, and placed in a 5 °C refrigerator for 12 h. Then, the bloods were centrifuged at 400× *g* at 4 °C for 15 min to obtain the serum. The serum was collected and transferred into a polypropylene tube and stored at −20 °C until the blood biochemical analysis.

The chronic intraperitoneally GnIH or saline-treated piglets were weighed and then killed by exsanguination after electrical stunning. Subsequently, the organs and tissues including the liver, spleen, kidney, pancreas, and inguinal subcutaneous white adipose tissue (iWAT) and perigonadal white adipose tissue (pgWAT) of the piglets were quickly harvested. The iWAT and pgWAT collection methods refer to the previous studies [50,51]. After all the samples were weighed on an electronic balance (C20002, WANTE, Shanghai, China), the liver, iWAT and pgWAT were frozen in a liquid nitrogen and stored at −80 °C for quantitative RT-PCR and Western blot, while the pancreas was fixed in neutral-buffered formalin for histology analysis.

### 4.6. Serum Biochemical Analyses

The total cholesterol (TC), triglycerides (TG), low-density lipoprotein (LDL-C), high-density lipoprotein (HDL-C), glucose (GLU), glutamic-pyruvic transaminase (ALT), glutamic-oxalocetic transaminase (AST) and lactic dehydrogenase (LDH) level in the serum was, respectively, detected by using Cholesterol Determination Kit (Cat. U82785040, Uritest, Guilin, China), Triglyceride Determination Kit (Cat. U82881050, Uritest, Guilin, China), Low Density Lipoprotein Cholesterol Determination Kit (Cat. U83085045, Uritest, Guilin, China), High Density Lipoprotein Cholesterol Determination Kit (Cat. U82985045, Uritest, Guilin, China), Glucose Determination Kit (Cat. U83781050, Uritest, Guilin, China), Alanine Aminotransferase Assay Kit (Cat. U80785040, Uritest, Guilin, China), Aspartate Aminotransferase Assay Kit (Cat. U80885040, Uritest, Guilin, China), Lactate Dehydrogenase Determination Kit (Cat. U84588060, Uritest, Guilin, China) on the automated biochemical analyzer (URIT- 8021avet, Uritest, Guilin, China).

### 4.7. Blood Glucose Measurements

After the fasting treatments, the blood glucose of the acute and chronic intraperitoneally GnIH or saline-injected piglets were immediately measured from the marginal ear vein blood samples at specific time points (15 min prior to and 0–60 min after an injection of different doses of GnIH or saline) using a blood glucose meter (FreeStyle Optium Neo, Abbott, Shanghai, China).

### 4.8. Glucose Tolerance Test

The glucose tolerance test was conducted as described in our previous study [29]. Before the glucose tolerance test, the chronic GnIH-injected piglets received intraperitoneal injections of different doses of GnIH twice a day for 14 days, as described above, whereas the rats of the acute treatment group were reared without any treatment. After fasting for 8 h, the piglets in acute and chronic treatment groups were all intraperitoneally injected with glucose at a dose of 1.5 g/kg body weight. Fifteen min later, different doses of GnIH or saline were intraperitoneally injected into the piglets. Subsequently, the blood glucose was immediately measured from the ear vein blood samples at specific time points, as described above.

### 4.9. Gene Expression

The total RNA was extracted using TRIzol reagent (Vazyme, Nanjing, China) for the liver, iWAT and pgWAT from chronic intraperitoneally GnIH-injected piglets. The first-strand cDNA was synthesized from the total RNA using a Reverse Transcription Kit (Cat. R211-02, Vazyme, Nanjing, China). Amplification reactions were conducted in triplicate using gene-specific primers designed from the clone sequences shown in Table 2. The PCR products in each cycle were monitored using a fluorescence quantitative PCR instrument (PRISMR 7500, ABI, Los Angeles, CA, USA). The quantitative RT-PCR was performed using the 2^−ΔΔCT^ method with β-actin as the internal control for the normalization.

### 4.10. Western Blot Analysis

The liver, iWAT and pgWAT from the chronic intraperitoneally GnIH-injected piglets were lysed in a cell lysis buffer (Beyotime, Shanghai, China) containing 1 mM of phenylmethylsulfonyl fluoride (Solarbio, Beijing, China). Western blotting was conducted as described in our previous study [52]. The blotted membranes were incubated with primary antibodies [GSK3β (Cat. D5C5Z, CST, Shanghai, China), Phospho-GSK3β (Cat. 5B3, CST, Shanghai, China), AKT (Cat. C67E7, CST, Shanghai, China), Phospho-AKT (Cat. D25E6, CST, Shanghai, China), AMPKα (Cat. 5831T, CST, Shanghai, China), Phospho-AMPKα (Cat. 2535T, CST, Shanghai, China), AMPKβ (Cat. 4150T, CST, Shanghai, China), Phospho-AMPKβ (Cat. 4186T, CST, Shanghai, China), C/EBPα (Cat. 8178T, CST, Shanghai, China), PPARγ (Cat. 2435T, CST, Shanghai, China), β-actin (Cat. 4970T, CST, Shanghai, China), ACLY (Cat. A3719, ABclone, Wuhan, China)] all at a 1:1000 dilution and were then incubated with horseradish peroxidase-labeled goat anti-rabbit (1:20,000 dilution, Cat. BA1054, CST, Shanghai, China). The densitometric quantification was performed with Image J(Version 1.53, NIH, Bethesda, MD, USA) using β-actin as the internal control for the normalization.

### 4.11. Morphological Analysis

The fixed pancreas was flushed with running water overnight. The tissue samples were dehydrated in a graded ethanol series and embedded in paraffin wax, followed by histological sectioning (5 mm). In the immunohistochemistry analysis, the tissue sections were incubated with a primary antibody at the appropriate dilution [insulin (Cat. BM0080) and glucagon (Cat. BA1621) (1:100 dilution, Boster Biological Technology, Wuhan, China)]. Immunohistochemical analyses were performed using streptavidin-biotin-peroxidase complex (SABC) kits (Cat. SA1021 and SA1022, Boster Biological Technology, Wuhan, China). After the primary antibody incubation, the sections were incubated with Cy3-conjugated goat anti-mouse IgG secondary antibodies (Cat. 13L24C, 1:200 dilution, Boster Biological Technology, Wuhan, China). After staining and blocking, microscopic photographs were taken, and the average optical density values were analyzed by image J.

### 4.12. Statistical Analyses

The statistical analysis was evaluated by unpaired two-tailed Student’s *t*-test or one-way ANOVA, followed by a post hoc Tukey’s test, with SPSS Statistics version 17.0. The differences were considered to be significant when *p* < 0.05. No data were excluded from the analyses.

## Figures and Tables

**Figure 1 ijms-23-13956-f001:**
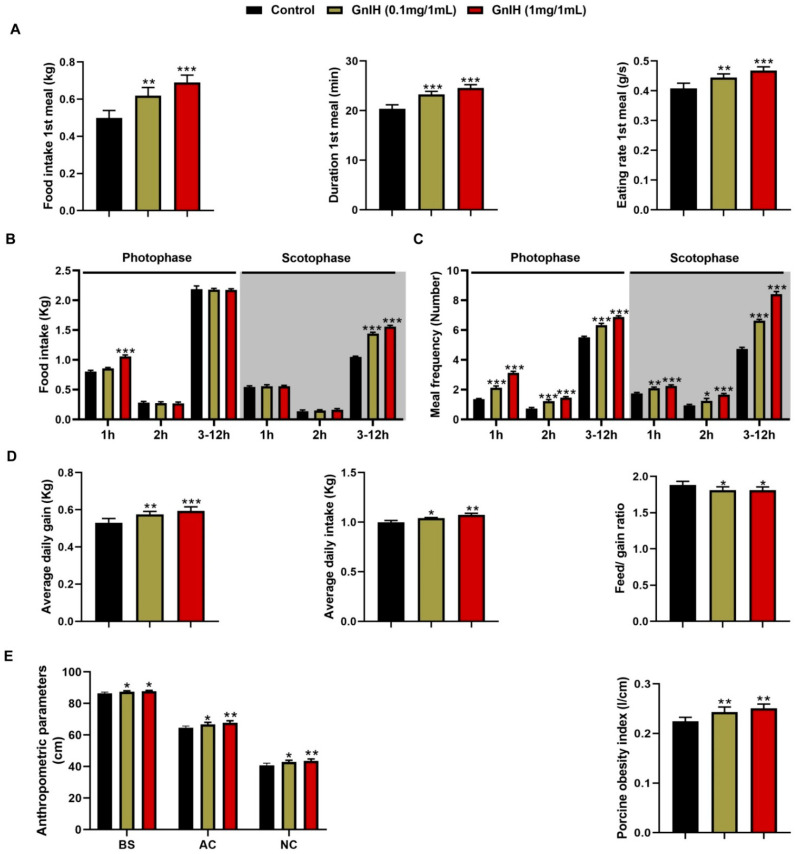
Intraperitoneally injected GnIH increases food intake, alters meal microstructure, decreases feed/gain ratio and increases body mass in piglets. (**A**) Effect of GnIH injected intraperitoneally on the structure of first meal in the fasting piglets. (**B**,**C**) The food intake (**B**) and meal frequency (**C**) were monitored over 14 days during different periods of the photophase and scotophase post-GnIH or vehicle injection in the ad libitum-fed piglets. (**D**) The average daily feed intake, average daily gain and feed/gain ratio of piglets intraperitoneally injected with different chronic doses of GnIH. (**E**) Anthropometric parameters and porcine obesity index of piglets intraperitoneally injected with different chronic doses of GnIH. BS, body size; AC, abdominal circumference; NC, neck circumference; POI, porcine obesity index. *n* = 5/group. The data are presented as the means ±SEM. * *p* < 0.05, ** *p* < 0.01 and *** *p* < 0.001 GnIH vs. vehicle.

**Figure 2 ijms-23-13956-f002:**
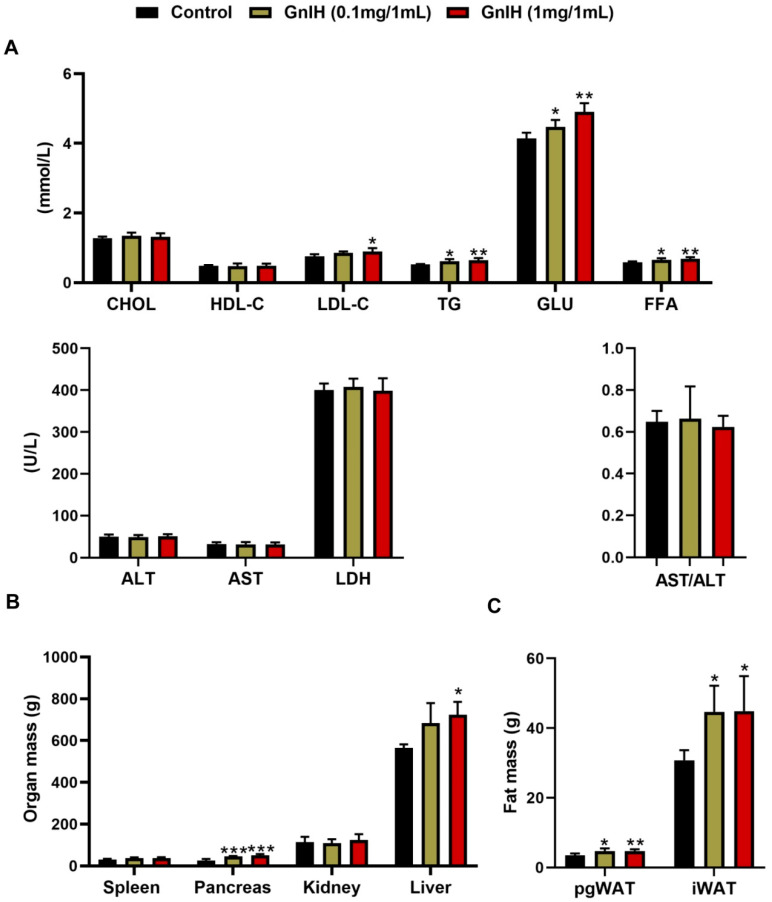
Intraperitoneally injected GnIH alters serum biochemical indexes and body fat composition. (**A**) The concentration of TG (total triglyceride), GLU (glucose), CHOL (cholesterol), HDL-C (high density lipoprotein cholesterol), LDL-C (low density lipoprotein cholesterin), LDH (lactic dehydrogenase), FFA (free fatty acids), ALT (alanine aminotransferase) and AST (aspartate aminotransferase) in the serum of chronic GnIH or vehicle-treated female piglets. (**B**) The organ mass and (**C**) fat mass in female piglets intraperitoneally injected with GnIH or vehicle for 14 days. *n* = 5/group. The data are presented as the means ± SEM. * *p* < 0.05, ** *p* < 0.01 and *** *p* < 0.001 GnIH vs. vehicle.

**Figure 3 ijms-23-13956-f003:**
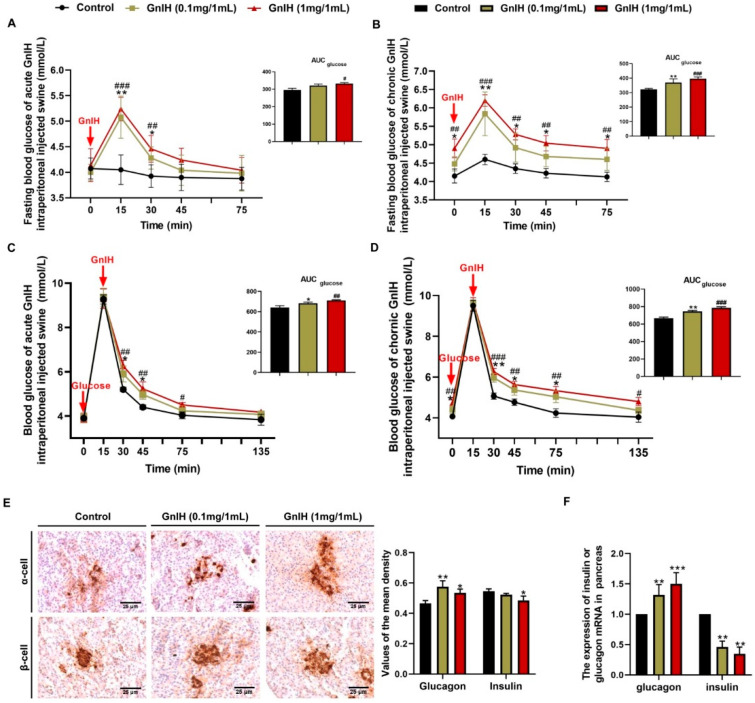
Effects of intraperitoneally injected GnIH on glucose homeostasis and islet histomorphology changes. (**A**,**B**) The fasting blood glucose levels of piglets were measured at different time points after intraperitoneally injecting different acute (**A**) or chronic (**B**) doses of GnIH or vehicle into rats that had fasted for 8 h. The upper panel shows the total area under the curve (AUC) for fasting blood glucose after GnIH or vehicle injection from 0 to 60 min. (**C**,**D**) For the intraperitoneal glucose tolerance test, blood glucose concentrations were measured in ad libitum-fed piglets that had been administered different acute (**C**) or chronic (**D**) doses of GnIH or vehicle. The upper panel shows the total AUC values for blood glucose after the administration of different doses of GnIH or vehicle from 0 to 120 min. (**E**) Representative immunohistochemistry images and mean densitometric values for glucagon or insulin in islets of pancreatic sections from piglets administered different chronic doses of GnIH or vehicle for 14 days. Scare bars: 25 μm. (**F**) *Glucagon*- and *insulin*-related gene expression in the pancreases of piglets administered different chronic doses of GnIH or vehicle for 14 days. *n* = 5/group. The data are presented as the means ±SEM. * *p* < 0.05, ** *p* < 0.01 and *** *p* < 0.001 GnIH 0.1 mg/1 mL vs. vehicle; # *p* < 0.05, ## *p* < 0.01 and ### *p* < 0.001 GnIH 1 mg/1 mL vs. vehicle.

**Figure 4 ijms-23-13956-f004:**
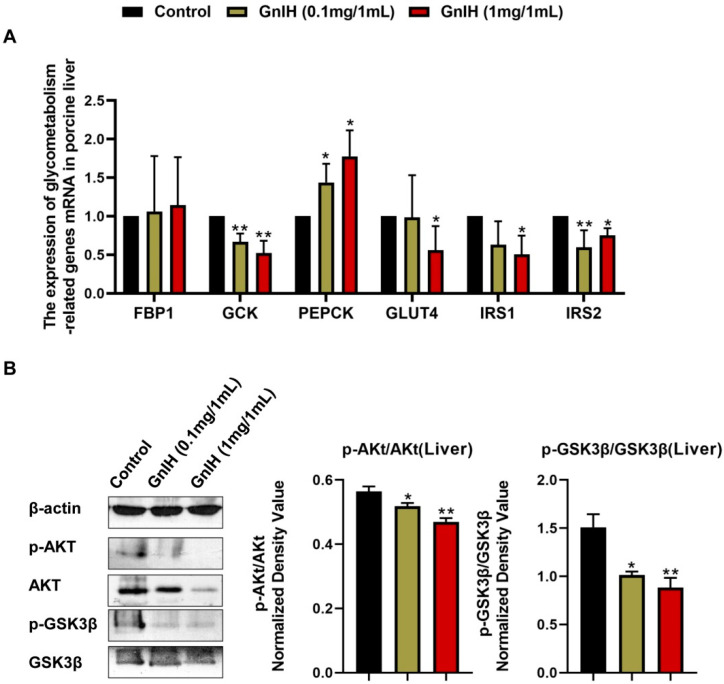
(**A**) The mRNA expression levels of hepatic glucometabolism-related genes in intraperitoneally GnIH-injected piglets. *GCK* as a key glycolysis enzyme, *PEPCK* and *FBP1* as critical gluconeogenesis enzymes, *GLUT4* as a key glucose transporter, *IRS1* and *IRS2* as insulin transduction factors were selected to detect by relative RT-PCR. (**B**) Effect of intraperitoneally injected GnIH on AKT-GSK3β signaling in the liver of piglets administered different chronic doses of GnIH or vehicle. Densitometric quantification of p-AKT and p-GSK-3β levels was performed after normalization to total AKT and GSK-3β levels as loading controls, respectively. *n* = 3 independent experiments. The data are shown as the means ±SEM. * *p* < 0.05 and ** *p* < 0.01. GnIH vs. vehicle.

**Figure 5 ijms-23-13956-f005:**
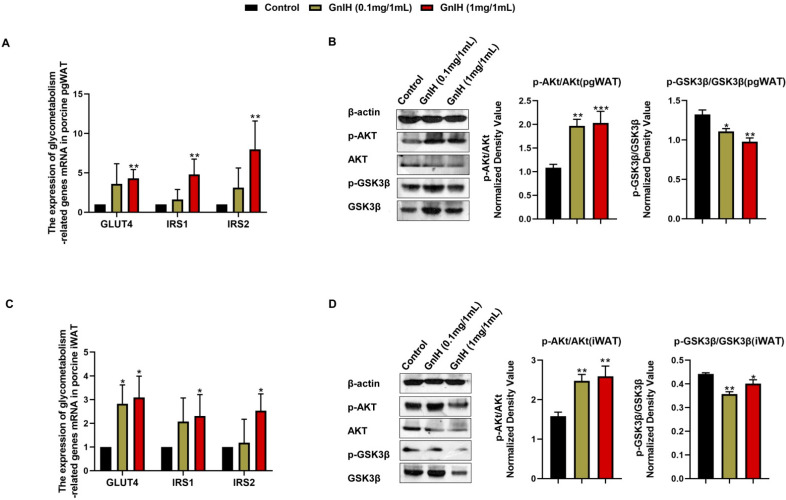
The mRNA expression levels of insulin resistance-related genes in pgWAT (**A**) and iWAT (**C**) of piglets administered different chronic doses of GnIH or vehicle for 14 days. Effect of intraperitoneally injected GnIH on AKT-GSK-related signaling in pgWAT (**B**) and iWAT (**D**) of piglets administered different chronic doses of GnIH or vehicle for 14 days. Densitometric quantification of p-AKT and p-GSK-3β levels was performed after normalization to total AKT and GSK-3β levels as loading controls. *n* = 3 independent experiments. pgWAT, perigonadal white adipose tissue; iWAT, inguinal white adipose tissue. The data are shown as the means ± SEM. * *p* < 0.05, ** *p* < 0.01 and *** *p* < 0.001 GnIH vs. vehicle.

**Figure 6 ijms-23-13956-f006:**
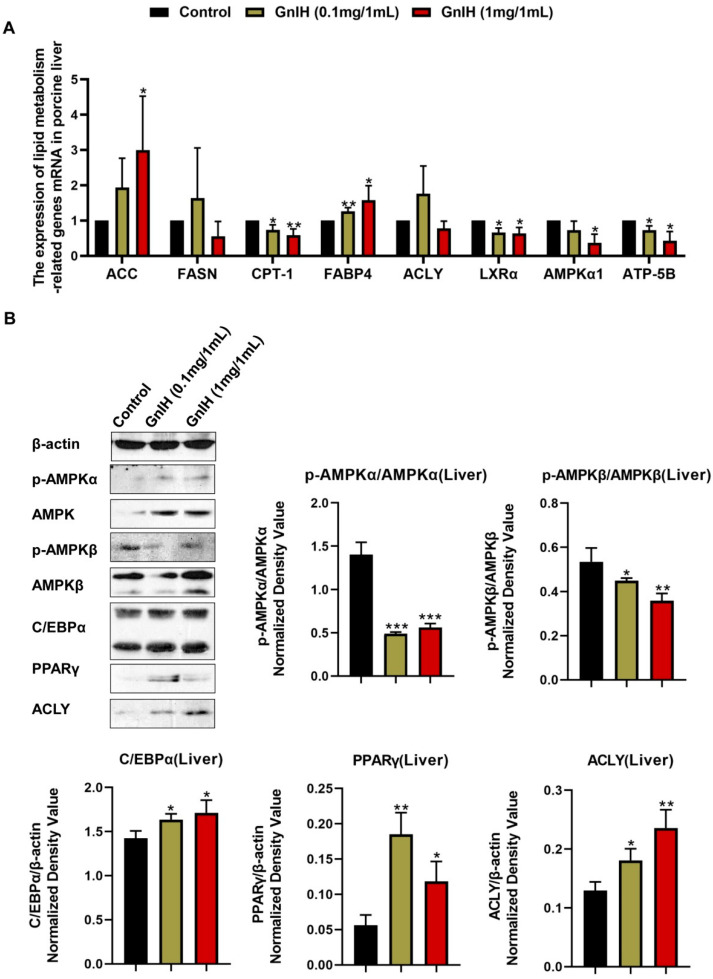
(**A**) The mRNA expression levels of hepatic lipid metabolism-related genes in liver of piglets administered different chronic doses of GnIH or vehicle for 14 days. *ACC*, *FASN* and *ACLY* as key lipogenic enzymes, *CPT-1* as critical lipolytic enzyme, *LXRα* as main cholesterol metabolizing enzyme, *FABP4* as a key lipid transporter, *AMPKα1* and *ATP-5B* as critical energy metabolism genes were selected to detect by relative RT-PCR. (**B**) Effect of intraperitoneally injected GnIH on AMPK and downstream signaling cascade in the liver of piglets administered different chronic doses of GnIH or vehicle for 14 days. Densitometric quantification of p-AMPKα and p-AMPKβ levels was performed after normalization to total AMPKα and AMPKβ levels as loading controls while the densitometric quantification of C/EBPα, PPARγ and ACLY was performed after normalization to β-actin levels as loading controls, respectively. *n* = 3 independent experiments. The data are shown as the means ±SEM. * *p* < 0.05, ** *p* < 0.01 and *** *p* < 0.001 GnIH vs. vehicle.

**Figure 7 ijms-23-13956-f007:**
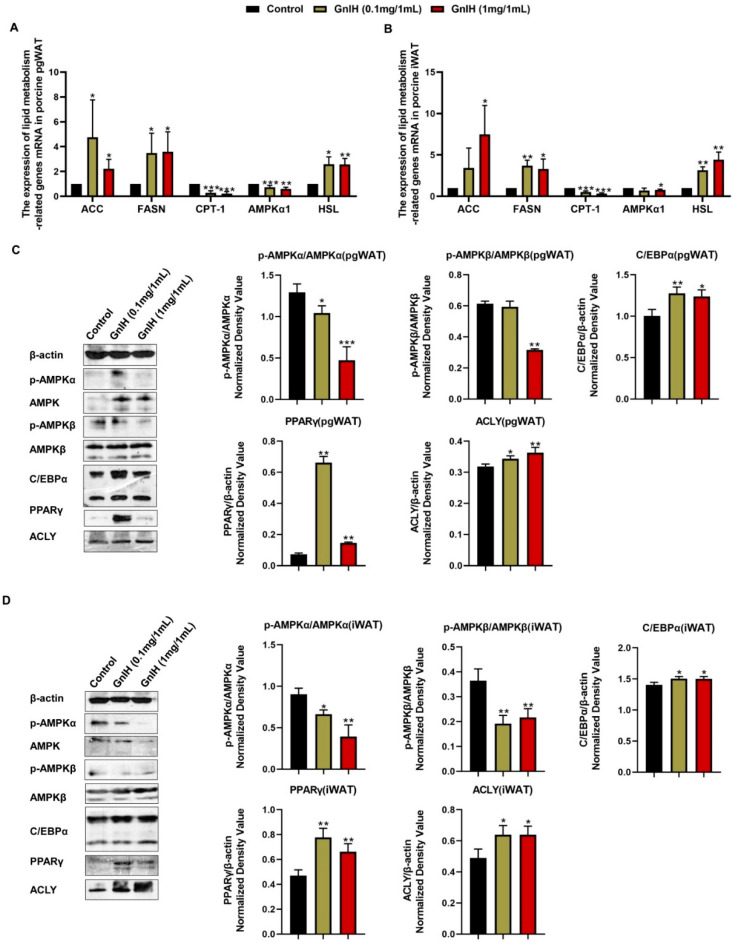
The mRNA expression levels of hepatic lipid metabolism-related genes in pgWAT (**A**) and iWAT (**B**) of piglets administered different chronic doses of GnIH or vehicle for 14 days. (**C**,**D**) Effect of intraperitoneally injected GnIH on AMPK and downstream signaling cascade in the pgWAT (**C**) and iWAT (**D**) of piglets administered different chronic doses of GnIH or vehicle for 14 days. Densitometric quantification of p-AMPKα and p-AMPKβ levels was performed after normalization to total AMPKα and AMPKβ levels as loading controls while the densitometric quantification of C/EBPα, PPARγ and ACLY was performed after normalization to β-actin levels as loading controls, respectively. *n* = 3 independent experiments. The data are shown as the means ±SEM. * *p* < 0.05, ** *p* < 0.01 and *** *p* < 0.001 GnIH vs. vehicle.

**Figure 8 ijms-23-13956-f008:**
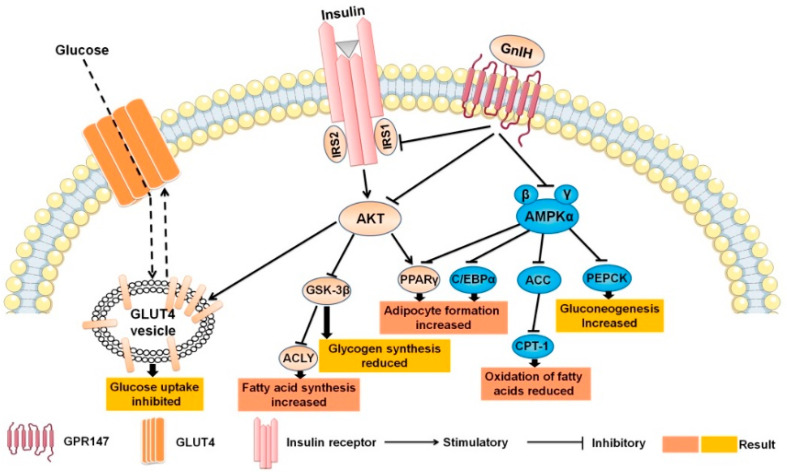
Effects of intraperitoneally injected GnIH on related mechanisms of glucolipid metabolism.

**Table 1 ijms-23-13956-t001:** Effect of GnIH injected intraperitoneally on meal microstructure in the ad libitum-fed piglets.

Parameters	Photophase (12 h)			Scotophase (12 h)		
	Control	GnIH	GnIH	Control	GnIH	GnIH
	(0 mg/1 mL)	(0.1 mg/1 mL)	(1 mg/1 mL)	(0 mg/1 ml)	(0.1 mg/1 mL)	(1 mg/1 mL)
Food intake (kg/12 h)	3.27 ± 0.02	3.31 ± 0.03	3.51 ± 0.02 ***	1.73 ± 0.02	2.17 ± 0.01 ***	2.27 ± 0.03 ***
Meal frequency (number/12 h)	7.59 ± 0.15	9.67 ± 0.20 ***	11.46 ± 0.19 ***	7.41 ± 0.13	9.99 ± 0.16 ***	12.3 ± 0.15 ***
Time spent in meals (min/12 h)	137.9 ± 4.87	151.5 ± 3.97 *	154.6 ± 1.85 **	121.8 ± 2.72	144.5 ± 2.27 ***	152.7 ± 1.77 ***
Meal size (kg/meal)	0.43 ± 0.01	0.34 ± 0.01 ***	0.30 ± 0.01 ***	0.23 ± 0.01	0.21 ± 0.01 *	0.18 ± 0.01 ***
Eating rate (g/s)	0.39 ± 0.02	0.37 ± 0.01	0.38 ± 0.01	0.24 ± 0.01	0.25 ± 0.01	0.25 ± 0.01
Meal duration (min/meal)	18.27 ± 0.29	15.67 ± 0.37 ***	13.5 ± 0.25 ***	16.40 ± 0.29	14.5 ± 0.29 ***	12.4 ± 0.22 ***
Inter-meal interval (min)	77.34 ± 1.32	58.79 ± 1.48 ***	49.34 ± 0.86 ***	80.67 ± 1.65	57.62 ± 1.02 ***	46.03 ± 0.55 ***
Satiety ratio (min/kg)	220.19 ± 1.46	217.32 ± 2.15	204.74 ± 0.99 ***	416.22 ± 3.93	331.30 ± 1.90 ***	316.78 ± 4.55 ***

Data are mean ± SEM; *n* = 5 piglets/group. * *p* < 0.05, ** *p* < 0.01 and *** *p* < 0.001 GnIH vs. vehicle.

**Table 2 ijms-23-13956-t002:** Primers and annealing temperature for relative real-time RT-PCR.

Gnens	Primer sequence (5′-3′)	Genebank No.	Function
*IRS1*	F:GAATCTCAGTCCCAACCGCAAC	NM_001244489.1	Insulin signal transduction
	R:CTGGGTGTCGAGGAGAAGGTCTC		
*IRS2*	F:ACAGACTAAATACAACGCACGACTC	XM_021065907.1	Insulin signal transduction
	R:GAAGTATATTTCTGGCTCTTGGGAC		
*LXRA*	F:ATCCGCCTGAAGAAACTGAAGC	XM_013994348.2	Cholesterol metabolism
	R:CTGGTCTGAAAAGGAGCGTCTG		
*FBP1*	F:CTCTCCAATGACCTGGTTATTAACG	NM_213979.1	Gluconeogenesis
	R:TTTCTGTAGATGCCAAAGATGGTTC		
*PEPCK*	F:TCAGCACGACTCCAGCCTTCA	NM_001123158.1	Gluconeogenesis
	R:GCTCAAGCAGTCTGGGCATTCT		
*GCK*	F:ATCAAACGGAGAGGGGACTT	XM_013985832.2	Glucose metabolism
	R:ACAATCATGCCAACCTCACA		
*FASN*	F:CTACGAGGCCATTGTGGACG	NM_001099930.1	Fatty acid synthesis
	R:AGCCTATCATGCTGTAGCCC		
*ACC*	F:AGCAAGGTCGAGACCGAAG	NM_001114269.1	Fatty acid synthesis
	R:TAAGACCACCGGCGGATAGA		
*ACLY*	F:GAGGCAGCATCGCAAACTTCA	XM_021066028.1	Fatty acid transport
	R:TCCCAACTTCTCCCATCACCC		
*ATP5B*	F:GAATCCCTTCTGCGGTGGGTTAT	XM_001929410.5	ATP synthase
	R:GGCAGGAGCAGGGTCAGTCAAGT		
*GlUT-4*	F:GTATGTTGCGGATGCTATGGG	NM_001128433.1	Glucose transporter
	R:CTCGGGTTTCAGGCACTTTTAG		
*CPT-1*	F:TCACAAGCGAATTTGAGTGC	NM_001129805.1	Fatty acid beta oxidation
	R:AAATTCAGACCGCAGTTTCG		
*FABP4*	F:AGTGGGATGGAAAGACAACCAC	NM_001002817.1	Fatty acid transport
	R:GTCGGGACAATACATCCAACAG		
*AMPKα1*	F:TGTCACAGGCATATGGTGGTC	XM_021076522.1	Energy metabolism
	R:GGACCAGCATACAACCTTCCT		
*Glucagon*	F:ACATTGCCAAACGTCACGATG	XM_005671883.3	Glucagon synthesis
	R:GCCTTCCTCGGCCTTTCA		
*Insulin*	F:GCCTTTGTGAACCAACACCTG	XM_021081278.1	Insulin synthesis
	R:GTTGCAGTAGTTCTCCAGCTG		
*β-actin*	F:TGGAACGGTGAAGGTGACAGC	XM_003124280.5	Reference genes
	R:GCTTTTGGGAAGGCAGGGACT		

## Data Availability

The data presented in this study are available on request from the corresponding author.

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
