# Peer review of "Peripheral Gonadotropin-Inhibitory Hormone (GnIH) Acting as a Novel Modulator Involved in Hyperphagia-Induced Obesity and Associated Disorders of Metabolism in an In Vivo Female Piglet Model"

_ijms, 2022, doi:10.3390/ijms232213956_

Round 1

Reviewer 1 Report

Dear authors, thank you for the study. As I see, you continued to study the mechanisms of action of gonadotropin-inhibitory hormone on various animal models. The present manuscript looks good, I have only minor concerns: please rewrite the Abstract, now it is hard to read and it does not correspond to the Journal's requirements.

Abstract: The abstract should be a total of about 200 words maximum. The abstract should be a single paragraph and should follow the style of structured abstracts, but without headings: 1) Background: Place the question addressed in a broad context and highlight the purpose of the study; 2) Methods: Describe briefly the main methods or treatments applied. Include any relevant preregistration numbers, and species and strains of any animals used. 3) Results: Summarize the article's main findings; and 4) Conclusion: Indicate the main conclusions or interpretations. 

Also, please add your view of future directions in research.

Reviewer 2 Report

- Why the authors decided to do the study on the pigs. I think that this decision was connected with the fact that the pig is good animal model for reactions in human organisms. Information about this fact should be added in the introduction. It is known about similarities between humans and pigs in organization of the nervous system and digestive tract. Are there such information about hormonal system?

- The novelty of the studies should be underlined at the end of the introduction.

- The first paragraph of discussion is about the aim of the study. I suggest to replace this paragraph to the introduction

- Conclusion at the end of discussion is too long and not clear. I suggest to show the conclusion in a few sentences with emphasis on research innovation.

- Whether the experiment had any limitations. please present them in the discussion

- The number the number and date of receipt of the approval of the ethics committee should be added to the materials and methods

- In lines 532-533 the authors wrote ”The concentrations of GnIH used in the present study were decided according to our pre-experiments and previous studies. In my opinion a short justification for the doses should be added with citation of references.

- Some references are rather old, published about 20 years ago. Some of them have been published in the 1990s. Are there no newer studies on this topic?
